# Effect of Pollyallylamine on Alcoholdehydrogenase Structure and Activity

**DOI:** 10.3390/polym12040832

**Published:** 2020-04-06

**Authors:** Aleksandr L. Kim, Egor V. Musin, Alexey V. Dubrovskii, Sergey A. Tikhonenko

**Affiliations:** Institute of Theoretical and Experimental Biophysics Russian Academy of Science, Institutskaya st., 3, Puschino 142290, Moscow Reg., Russia; kimerzent@gmail.com (A.L.K.); eglork@gmail.com (E.V.M.); dav198@mail.ru (A.V.D.)

**Keywords:** polyallylamine, alcohol dehydrogenase, NaCl, (NH_4_)_2_SO_4_, alcohol dehydrogenase structure, catalytic characteristics

## Abstract

In this article, the effect of polyallylamine (PAA) on the structure and catalytic characteristics of alcohol dehydrogenase (ADH) was studied. For this research, we used methods of stationary kinetics and fluorescence spectroscopy. It has been shown that PAA non-competitively inhibits ADH activity while preserving its quaternary structure. It was established that 0.1 M ammonium sulfate removes the inhibitory effect of PAA on ADH, which is explained by the binding of sulfate anion (NH_4_)_2_SO_4_ with polyallylamine amino groups. As a result, the rigidity of the polymer chain increases and the ability to bind to the active loop of the enzyme increases. It is also shown that sodium chloride removes the inhibitory effect of PAA on ADH due to an electrostatic screening of the enzyme from polyelectrolyte. The method of encapsulating ADH in polyelectrolyte microcapsules was adapted to the structure and properties of the enzyme molecule. It was found that the best for ADH is its encapsulation by adsorption into microcapsules already formed on CaCO_3_ particles. It was shown that the affinity constant of encapsulated alcohol dehydrogenase to the substrate is 1.7 times lower than that of the native enzyme. When studying the affinity constant of ADH in a complex with PAA to ethanol, the effect of noncompetitive inhibition of the enzyme by polyelectrolyte was observed.

## 1. Introduction

Currently, for the diagnosis of human diseases, clinical and biochemical methods of analysis are widely used. They are associated with the determination of substances in biological fluids, mainly in the blood, serum, plasma or urine. The following requirements are imposed on such methods: High sensitivity, accuracy, specificity, and high reaction rate. Enzymatic methods used in clinical diagnostics have the above-mentioned properties [1]; however, they have several disadvantages, namely, small shelf-life of enzymes in solution, the inability to use enzymes in the presence of proteinases, and a single use of the enzyme [2]. The encapsulation of the enzyme in polyelectrolyte microcapsules (PMCs) eliminates these disadvantages. PMCs are obtained by alternately adsorbing oppositely charged polyelectrolytes on a colloid particle, with its subsequent removal [3,4]. These microcapsules have a diameter of from 0.5 to 10 microns and a semi-permeable shell, the composition and thickness of which can be controlled [5,6,7].

The use of enzymes encapsulated in PMC has several advantages over the use of the native enzyme for clinical diagnostics use. Sukhorukov et al. showed that the encapsulated urease retains its activity for more than six months, while the native enzyme lasts only a week; in addition, the protein was not exposed to the negative effects of proteases in solution [8]. Currently, there are a number of works demonstrating these properties in diagnostic systems, for example, Reshetilov et al. proposed a biosensor for detecting urea based on a pH-sensitive field effect transistor, on which microcapsules containing urease are fixed [9]. Another work presents a similar biosensor, but with encapsulated glucose oxidase. At the same time, both detectors are reusable, since the enzyme sensor is protected from the effects of an aggressive environment and is immobilized in a PMC.

To create this type of diagnostic system, it is necessary to select polyelectrolytes that have a minimal impact on the encapsulated protein to encapsulate the enzyme with maximum preservation of their activity. In this regard, it is necessary to study the polyelectrolyte-protein interactions, as well as the search for ways to neutralize the effects of polymers on the encapsulated enzyme.

There are a number of works in the literature that solve similar problems and show various options for complexation and the effect of polyelectrolyte on protein [10,11,12]. Thus, in the work of Durdenko et al., the effect of inorganic monovalent and divalent salts on the complexation of the PAA polycation with the urease enzyme was studied. It was shown that in solutions of monovalent salts of NaCl, KCl and NH_4_Cl, polyelectrolyte-protein complexes are formed as a result of electrostatic interactions, which monotonically decrease with increasing salt concentration, while in solutions of divalent salts of Na_2_SO_4_ and (NH4)_2_SO_4_, the formation efficiency of polyelectrolyte-protein complexes changed (enzyme activated) at low salt concentrations [13].

Saburova et al. studied the effect of polyelectrolytes on the stability and catalytic characteristics of pig lactate dehydrogenase (LDH) and bovine liver glutamate dehydrogenase (GDH), and it was shown that the binding of negatively charged polyelectrolytes—polystyrene sulfonate, polymethacrylate and polyphosphate—destroys the tertiary and partially the secondary structure of LDH and GDH, leading to their complete inactivation at pH < 7. For the concentration of polyelectrolytes required in this case, the inhibition of enzymes was two or more orders of magnitude lower than the corresponding concentrations for monomers. It was proposed that the oligomeric state of enzymes causes polyelectrolytes to act on them in a special way; this particular effect is significantly different from the effect of polyelectrolytes on monomeric enzymes. The effect is that polyelectrolytes break down the oligomeric structure of enzymes, and when this “cleavage" effect is greater, the hydrophobicity of the polyelectrolyte chain increases [14].

In particular, the literature describes experiments with synthetic and natural linear polyelectrolytes, such as heparin [15], hyaluronic acid [16], polyallylamine hydrochloride (PAA) [17] and polydiallyldimethylammonium chloride (PDADMAC) [18], when they interact with various proteins carried out by titration and modeling. Heparin has been shown to prevent and reverse native aggregation of bovine serum albumin (BSA), β-lactoglobulin (BLG) and Zn-insulin at a pH around pI and at low ionic strength. In addition, at pH values around pI, hyaluronic acid has a greater affinity for BSA than for BLG. A similar effect was observed in other works with polycations (PDADMAC, PAA); a change in protein selectivity was shown due to a change in the charge of polyelectrolyte. Stronger binding of both proteins (BSA, BLG) with polycations may be associated with higher chain flexibility and effective linear charge density of these polycations.

Additionally, Kayitmazer et al. [19] studied the critical conditions of complexation and coacervation for PDADMAC–BSA pairs by means of turbidimetric titration and found that the state of coacervation may be related to pH, ionic strength, and stoichiometry. Yu et al. [20] conducted a comprehensive study of the binding of human serum albumin with polyacrylic acid and associated this process with a heterogeneous distribution of the charge of the protein. da Silva et al. showed repulsion between complexes with the same charge between spherical polyelectrolyte brushes and β-lactoglobulin (BLG) [21].

Polyelectrolytes, both synthesized and natural, were used to stabilize the protein and prevent its aggregation. For example, Xu et al. used heparin to reverse and inhibit the aggregation of three proteins—BSA, BLG and Zn-insulin—while maintaining the original protein structure [15]. According to the results of turbidimetric titration and DLS, the aggregation of these proteins, both in the native and denatured state, can be well controlled by the formation of soluble complexes with heparin. Furthermore, Seyrek et al. [17] studied the interaction of BSA with hydrophobically modified polyacrylic acid and showed that the maximum affinity in the polyelectrolyte–protein complex was observed at ionic strength, at which the Debye radius is equal to the radius of the protein. Thus, inhomogeneous distributions of the Coulomb potential on the protein surface may play a role in tuning the overall affinity due to interactions with the polymer segments, which are not strictly related but are in close proximity to the protein.

In addition, Dubrovsky et al. [22] investigated the effect of PSS on the catalytic activity of alcohol dehydrogenase, and it was shown that this polymer inhibited the activity of the enzyme, but the use of 0.1 M ammonium sulfate removed this effect by competitively binding the SO_4_^2−^ ions to the protein molecule.

The purpose of this work is to study the effect of polyallylamine on the activity and structure of alcohol dehydrogenase in the presence of salts. This study enables the creation of a diagnostic system of multiple-use based on polyelectrolyte microcapsules with encapsulated alcohol dehydrogenase for measurement concentration of ethanol. 

## 2. Materials

The polyelectrolytes of polystyrene sulfonate (PSS) and polyallylamine (PAA) 70 kDa, yeast acohol dehydrogenase (ADH1, UniProtKB - P00330), sodium chloride and ammonium sulfate (99% purity) (Reachim, Moscow, Russia) were used.

### 2.1. Registration of the Kinetics of Enzymatic Reactions 

The kinetics of the enzymatic reactions was registered according to a change in the optical density in the NADH absorption band of 340 nm with the use of a Cary 100 spectrophotometer (Agilent, Harbor City, CA, United State). The reaction was initiated by the addition of 100 μL of the enzyme solution to 1.9 μL of the reaction mixture containing 0.1 mM ethanol, 0.2 mM NAD^+^ in 0.05 M Tris-HCl (pH 7.2). To study the PE effect on the enzyme, the former was added in the ratio of 1:1 and 1:5 (wt/wt). The influence of the inorganic ions on the PE–protein interaction was studied in the presence of 2.0 M NaCl, 0.2 M NaCl, and 0.1 M (NH_4_)_2_SO_4_. The salt solutions included PEs but excluded the enzyme, which was added prior to incubation. 

### 2.2. Determination of ADH Activity 

The ADH activity was determined as the ratio of the change in the optical density (A) and time period (t) according to a slope of the linear part of the NADH accumulation curve over 10 s from the start of the reaction. The concentration of the NAD coenzyme (0.2 mM) was significantly higher than the Km value (coenzyme saturation). For each measurement, the average value and standard deviation were obtained. 

### 2.3. Determination of Fluorescence Intensity of ADH

ADH fluorescence spectra were registered with the use of a Cary Eclipse unit (Agilent, Harbor City, CA, United States) in a 1 cm thermostatic cuvette (20 °C) and excitation at 273 nm and emission at 340 nm. The reaction mixture consisted of 50 μg/mL ADH, 50 μg/mL PSS, and 0.05 M Tris-HCl (pH 7.2). For each measurement, the average value and standard deviation were obtained.

### 2.4. Preparation of PMCs

At the first stage, CaCO_3_ microspherolites were obtained; for this, 0.33 M Na_2_CO_3_ was added to a vigorously stirred 0.33 M solution of CaCl_2_ of equal volume. The resulting suspension was precipitated, and the supernatant was decanted and used to produce PLA.

The shell of polyelectrolyte microcapsules on CaCO_3_ microspherolites was formed by their alternate incubation in solutions of the PAA polycation and the PSS polyanion with a concentration of 2 mg/ml containing 0.5 M NaCl. After each incubation, the samples were washed three times with 0.5 M NaCl to remove non-adsorbed polymer molecules. After applying the required number of layers, the carbonate microparticle was dissolved and removed by incubation in a 0.2 M EDTA solution for 2 h. 

### 2.5. Adsorption Protein Encapsulation

The microcapsules obtained in Section 2.4 were incubated in a protein solution with a concentration of 6 mg/ml for 12 h. Due to its porous structure, the protein was adsorbed into a microcapsule. The obtained microcapsules were washed three times with double-distilled water to remove non-adsorbed protein molecules.

### 2.6. Statistical Analysis of Data

Each measurement of determination of fluorescence intensity of ADH and ADH activity, the average value and standard deviation were obtained. The number of repeats (N) was 5. The significance of differences was tested using independent two-sample t-test (Student’s t-test), *p* ≤ 0.01. 

## 3. Results

### 3.1. Study of the Interaction of Polyallylamine with Alcohol Dehydrogenase

We studied the effect of PAA on ADH, and for that reason different polyelectrolyte concentrations were chosen equal in weight to the enzyme and five times higher than that value. Those concentrations of the enzyme were chosen because the concentration of polyelectrolyte was equal to the concentration of enzyme during preparation of the polyelectrolyte microcapsule. A five-fold increase in the concentration of polyelectrolyte will allow us to see more pronounced effects associated with the effect of the polymer on ADH.

Figure 1 shows the dependence of the activity of native ADH and in combination with PAA on the time of incubation. PAA inhibits the activity of the enzyme in the first minutes of incubation (paired t-test for the first point: ADH:PAA 1:1 = 21, ADH:PAA 1:1 = 20.3; *p* ≤ 0.01; paired t-test for the last point: ADH:PAA 1:1 = 8, ADH:PAA 1:1 = 8; *p* ≤ 0.01). In the future, there is no effect of polyelectrolyte on the activity of alcohol dehydrogenase; the decrease in activity over time is due to inactivation of the enzyme. In Figure 1, the ADH-PAA complex has a significantly lower activity during the entire incubation period compared to the native enzyme.

Since the decrease in enzyme activity can be associated both with a change in its structure and with the formation of a complex with a polymer, the task was to study the effect of a polymer on the structure of ADH using fluorescence spectroscopy. The tyrosine, phenylalanine and tryptophan amino acids are located both inside the globule and on its surface. Thus, if the structure of the enzyme is disturbed, it will lead to the increase of solvent influence on tryptophan, which was buried deep in a hydrophobic environment, and the fluorescence quantum yield would also decrease [23]. 

Figure 2 shows the dependence of the intrinsic fluorescence of alcohol dehydrogenase on incubation time, both native and in the presence of polyelectrolyte. It can be seen from Figure 2 that the presence of PAA in the solution does not affect the fluorescence of the protein both at the initial time point and during prolonged incubation (paired t-test for the first point: ADH:PAA 1:1 = 0,3, ADH:PAA 1:1 = 2.6; *p* ≤ 0.01). From this we can conclude that polyelectrolyte does not destroy the structure of the protein, because if the fluorescence of alcohol dehydrogenase does not decrease, then the quaternary structure is not disturbed [24].

It was found that PAA affects the activity of the enzyme but does not affect its structure. We studied the mechanisms of enzyme inhibition to understand this effect. Considering the electrostatic nature of the formation of the protein–polyelectrolyte complex, ammonium sulfate and sodium chloride salts were used by way of example of bivalent and monovalent salts. In the work of Tikhonenko et al. [13] it was shown that these salts affect the binding mechanism of cationic polyelectrolyte polyallylamine hydrochloride (PAA) with the oligomeric urease enzyme; in particular, the presence of 0.005 M ammonium sulfate or 0.2 M sodium chloride stabilized the enzyme and partially removed the negative effect of PAA.

The effect of sodium chloride and ammonium sulfate on the activity of ADH in the presence of PAA was studied. In Figure 3a, the dependence of ADH activity in the native state and in the presence of PAA in a ratio of 1:1 and 1:5 on the concentration of sodium chloride is shown, from which ADH is inhibited in the presence of a polymer. The inhibition of the enzyme by polyelectrolyte decreased when NaCl was added to the ADH solution in the presence of PAA. The enzyme activity was restored to the value of the activity of the native enzyme with increasing salt concentration, up to 0.05 M sodium chloride. This effect can be explained by the fact that salt counterions shielded charges on both the polyelectrolyte and the enzyme, which led to a decrease in the interaction between them. After this, there is a gradual decrease in the activity of ADH both in the native state and in the presence of PAA, while their trends do not differ. Such a drop in protein activity is associated with a decrease in its solubility due to salting out and dehydration with salt [25].

Figure 3b shows the dependence of the activity of ADH in the native state and in the presence of PAA in a weight ratio of 1:1 and 1:5 of the concentration of ammonium sulfate, from which it is clear that ADH is inhibited in the presence of a polymer. When (NH_4_)_2_SO_4_ was added to the ADH solution in the presence of PAA, the inhibition of the enzyme by polyelectrolyte decreased. The enzyme activity was restored to the value of the activity of the native enzyme with increasing salt concentration, up to 0.005 M ammonium sulfate. In this case, the concentration of ammonium sulfate is not sufficient for a shielding effect between the polyelectrolyte and the enzyme to occur, therefore preventing inhibition of the enzyme may be due to the fact that the sulfate anion of the salt binds to two amino groups of the polyelectrolyte, which results in an increase in the polymer chain stiffness and a decrease in its degrees of freedom and, consequently, to reduce the polyelectrolyte–protein interaction. This allows us to conclude that the decrease in activity of alcohol dehydrogenase in the presence of polyelectrolyte is presumably associated with the blocking of the active loop of the enzyme during the formation of the protein–polymer complex. Following this, a gradual decrease in the activity of ADH is observed both in the native state and in the presence of PAA, while their activities do not differ. The decrease in protein activity is associated with salting out and dehydration with salt. As a result, it can be seen that the optimal concentration of (NH_4_)_2_SO_4_ for removing inhibition of the enzyme by polyelectrolyte (PAA) is in the range of from 0.005 M to 0.15 M.

Since it is supposed to include ADH in PMC, which are stored for a long time in solution, we studied the effect of 0.15 M (NH_4_)_2_SO_4_ on the activity of ADH in the presence of PAA over time. This concentration of ammonium sulfate was chosen to show the effect of two mechanisms of polyelectrolyte–protein interaction: The ionic strength of 150 mmol of ammonium sulfate corresponds to the ionic strength of 300 mmol of sodium chloride, at which a decrease in enzyme inhibition was observed; and the effect of increasing the rigidity of the polymer chain sulfate ions was also taken into account. Figure 4 shows the dependence of the activity of native ADH in the presence of PAA in a ratio of 1:1 and 1:5 at the time of incubation in 0.15 M ammonium sulfate. From Figure 4, it can be seen that (NH_4_)_2_SO_4_ removes the inhibition of the enzyme polyallylamine. Upon further incubation of the solution, there is a gradual decrease in the activity of both native ADH and in combination with PAA. Thus, we see that 0.15 M ammonium sulfate removes the inhibition of the enzyme by polyelectrolyte during the entire incubation time.

Considering that PSS does not affect the activity of ADH [6], and PAA inhibits the enzyme, it is recommended to use 5 mM ammonium sulfate during storage and use of ADH containing microcapsules, since the presence of this salt removes the inhibitory effect of PAA on ADH. 

### 3.2. Encapsulation of ADH in Polyelectrolyte Microcapsules

The results obtained on the effect of PSS and PAA on the structure and activity of alcohol dehydrogenase allowed us to develop polyelectrolyte microcapsules, the shell of which contained polyelectrolytes minimally affecting ADH.

Based on the fact that prolonged incubation of ADH in EDTA solution leads to enzyme inactivation, which is associated with the loss of Zn^2 +^ from the protein molecule due to the formation of a complex between EDTA and zinc ions which are necessary for stabilizing the quaternary structure of alcohol dehydrogenase [26], the application of protein encapsulation techniques as described in Volodkin et al. [27] is not suitable. In this context, it was proposed to encapsulate the enzyme by the method of adsorption [28].

A distinctive feature of this technique is the preparation of microcapsules without the inclusion of protein in the CaCO_3_ core in the first stage. The polyelectrolyte microcapsules formed on the CaCO_3_ microspherolyte containing no protein have a porous intra-volume structure [29]. As a result, the resulting PMC adsorbed 30% of protein from solution. 

### 3.3. Comparison of Alcohol Dehydrogenase Affinity in Complex with Polyallylamine and in a Polyelectrolyte Microcapsule

The reaction rates catalyzed by the enzyme were investigated depending on the concentration of ethanol to study the affinity of native ADH, in combination with PAA and in an encapsulated state to the substrate. Figure 5 shows the curves of the reaction rate of the oxidation of ethanol, catalyzed by ADH, on its concentration. As can be seen from the Figure 5a, the affinity constant of encapsulated alcohol dehydrogenase to the substrate is 1.7 times lower than that of the native enzyme.

The Michaelis constant Km for ADH in the PMC was 3.3 mg/ml, while for native ADH this value was 1.9 mg/ml; V_max_ for encapsulated ADH was 30% of the value of V_max_ of the native enzyme. This effect may be a result of the enzyme being immobilized in a polyelectrolyte microcapsule.

From Figure 5b, the Michaelis constant of alcohol dehydrogenase in complex with PAA is 3.4 times lower than that of the native enzyme. The Michaelis constant Km of native ADH was 0.38 mg/ml, and for ADH in complex with PAA it was 0.11 mg/ml; V_max_ for ADH in complex with PAA was 41% of the value of V_max_ of the native enzyme. However, the resulting affinity constant is false, since there is a persistence of an increase in activity with increasing substrate concentration to 1 mg/ml, where the enzyme activity in a complex with PAA reaches a plateau with a further increase in polymer concentration. This effect is due to non-competitive inhibition of the enzyme.

The limitations of this study are connected with salt hydrolysis. Salts with divalent anions, except sulfates, undergo hydrolysis, and then it is not possible to conduct this experiment with reliable results. However, the mechanism of interaction of the sulfate anion with PAA that we proposed does not exclude the same result with other divalent salts. To deal with this limitation, we plan to make a mathematical model and to identify the influence of salt valence for the enzyme–PAA interaction.

## 4. Conclusions 

This work shows the inhibition of the activity of the enzyme polyallylamine, which was observed in the first minutes of incubation, while the structure of the enzyme was preserved.

This inhibition of ADH by polyelectrolyte is removed in the presence of sodium chloride and ammonium sulfate. The effect of ammonium sulfate and sodium chloride on the binding mechanism of cationic polyelectrolyte polyallylamine hydrochloride (PAA) with the oligomeric enzyme ADH was studied. It is shown that the formation of polyelectrolyte–protein complexes is of an electrostatic nature, and the corresponding interaction weakens with an increase in salt concentration of more than 100 mM according to the classical law of statistical physics, connecting the Debye radius with the ionic strength of the solution. In (NH_4_)_2_SO_4_ solutions, a sharp decrease in enzyme inhibition is observed at low salt concentrations from 5 mM, which does not fit into the framework of the classical theory of interaction of charges in solutions with a different ionic strength. In this case, a decrease in enzyme inhibition may be due to the fact that the sulfate anion of the salt binds to two amino groups of the polyelectrolyte, which results in an increase in the chain stiffness of the polymer and a decrease in its degrees of freedom and, as a result, a decrease in the polyelectrolyte–protein interaction. 

It was proposed to encapsulate ADH by adsorption in pre-prepared polyelectrolyte microcapsules consisting of polyallylamine (PAA) and polystyrenesulfonate (PSS), given the presence of divalent zinc in the structure of ADH.

The catalytic characteristics of ADH in a native state, in combination with PAA and encapsulated in polyelectrolyte microcapsules, were studied. The affinity constant of the encapsulated alcohol dehydrogenase to the substrate is 1.7 times lower than that of the native enzyme, which may be due to the immobilization of the enzyme. When studying the affinity constant of ADH in a complex with PAA to alcohol, the effect of noncompetitive inhibition of the enzyme by polyelectrolyte was observed.

## Figures and Tables

**Figure 1 polymers-12-00832-f001:**
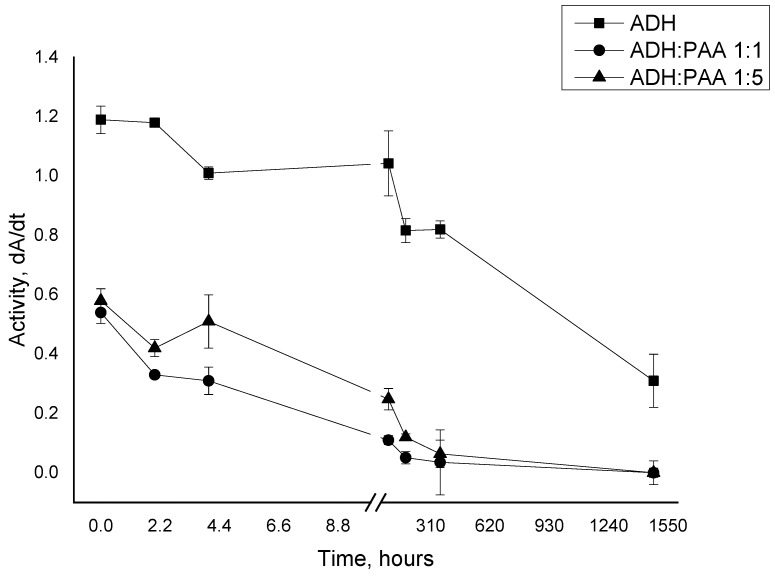
The dependence of the activity of ADH (alcohol dehydrogenase) (6.25 µg/ml) in the native state and in the presence of PAA (6.25 µg/ml, 1:1; 31.25 µg/ml, 1:5) from the time of incubation.

**Figure 2 polymers-12-00832-f002:**
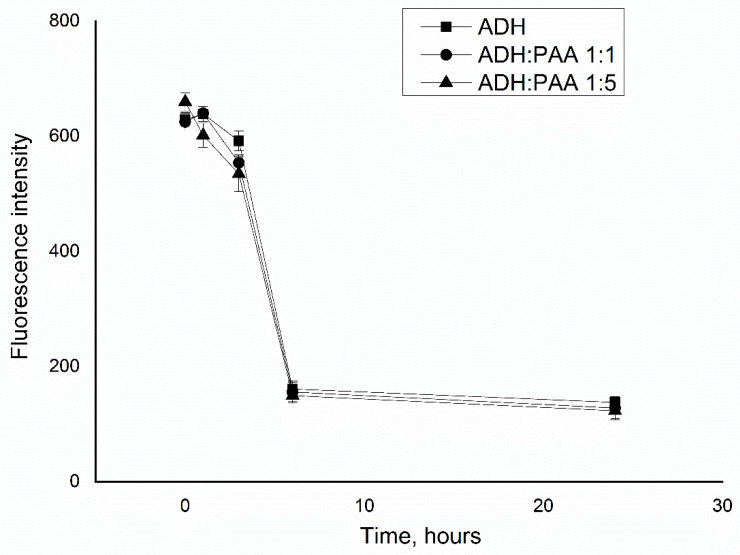
Dependence of the fluorescence of ADH (31.25 μg/ml) in the native state and in the presence of PAA (31.25 μg/ml, 1:1; 156.25 μg/ml, 1:5) from the time of incubation.

**Figure 3 polymers-12-00832-f003:**
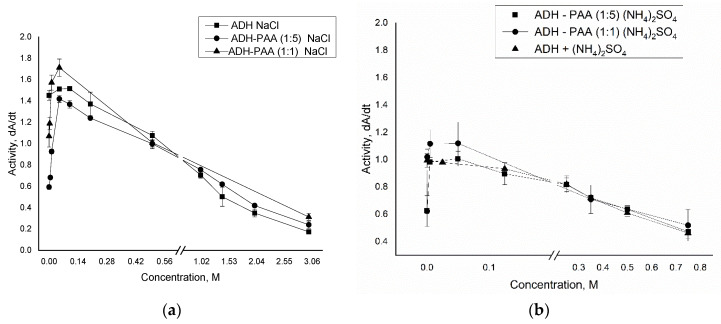
Dependence of ADH activity (6.25 µg/ml) in the native state and in the presence of PAA (6.25 µg/ml, 1:1; 31.25 µg/ml, 1:5) on the concentration of sodium chloride (**a**) and ammonium sulfate (**b**).

**Figure 4 polymers-12-00832-f004:**
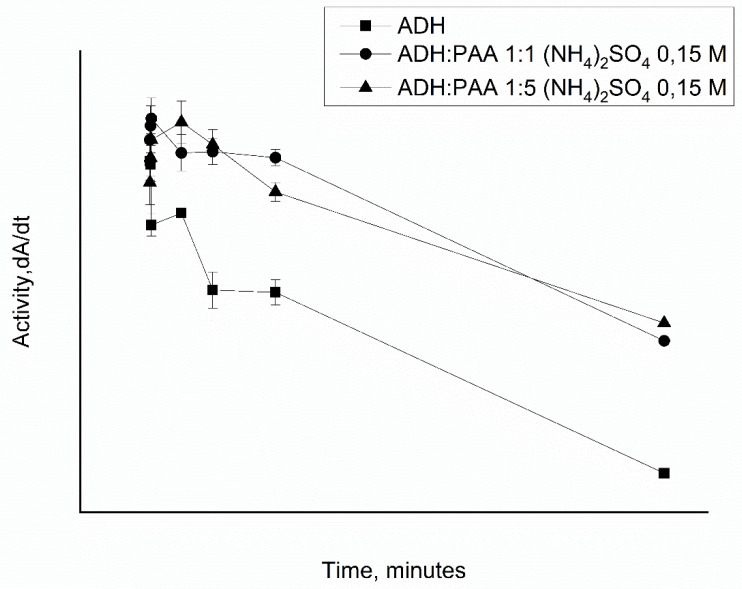
The dependence of the activity of ADH (6.25 μg/ml) in the native state and in the presence of PAA (6.25 μg/ml, 1:1; 31.25 μg/ml, 1:5) and ammonium sulfate from the time of incubation.

**Figure 5 polymers-12-00832-f005:**
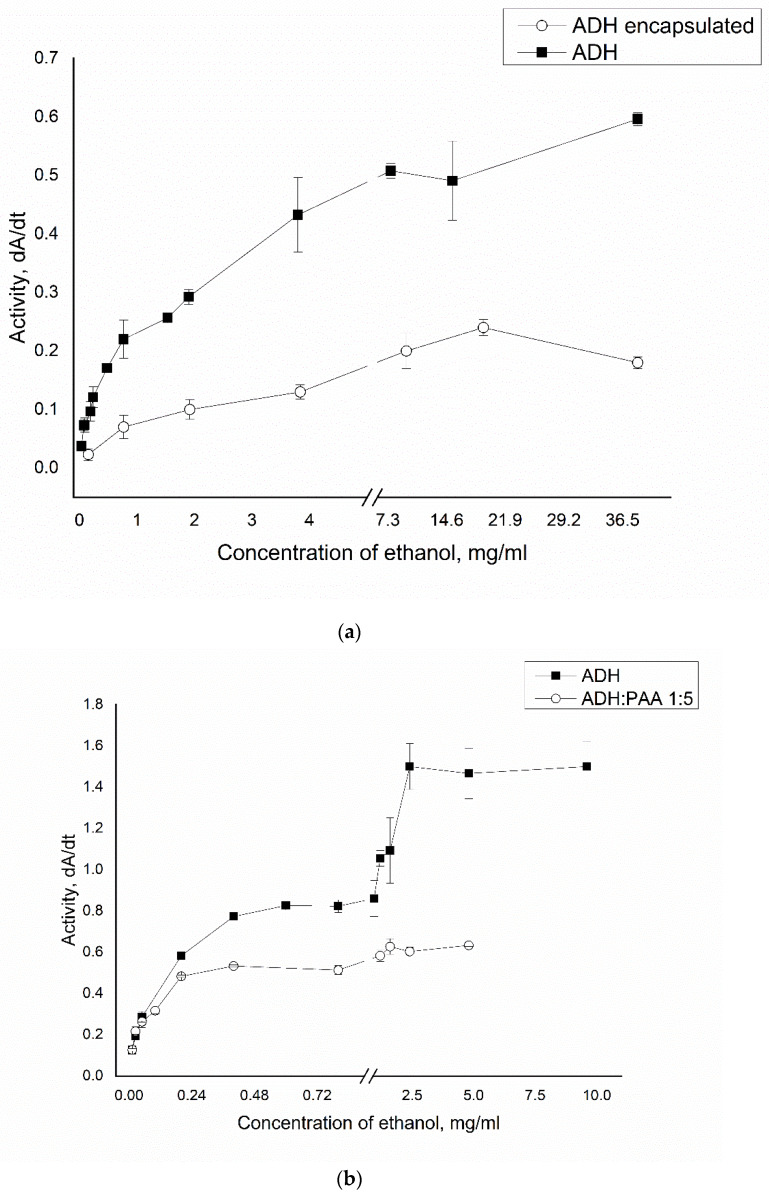
The dependence of the activity of ADH in the native state, in combination with PAA (**b**) and in capsules (**a**) on the concentration of ethanol.

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
