# Peer review of "Effect of Pollyallylamine on Alcoholdehydrogenase Structure and Activity"

_polymers, 2020, doi:10.3390/polym12040832_

Round 1

Reviewer 1 Report

Study of the Effect of Pollyallylamine on Alcoholdehydrogenase for Its Encapsulation into Polyelectrolyte Microcapsules

In this work, Kim et al. describe a study on the stabilizing effects of polyallylamine (PAH) on alcohol dehydrogenase (ADH) enzymes into polyelectrolyte microcapsules. The authors systematically investigated the impact of the PAH polycation on ADH, focusing on (1) non-competitive inhibition assays with NaCl and (NH3)2SO4, (2) microcapsule formation with CaCO3 particles, and (3) affinity constant of ADH with PAA to ethanol.

Overall, this study demonstrates an interesting approach to improve fundamental understanding of charged polymer/enzyme stabilization. The authors provide promising evidence that this is an effective way to preserve enzymatic activity for bioapplications. However, in its current state, the manuscript is still in an incomplete state for publication. Additional control experiments and characterization tools are needed to ensure that the conclusions are sound so that the materials can be reproduced. Considerable revisions are needed for this to be considered for Polymers, and publication at this time is premature. I encourage the authors to address these concerns and resubmit this manuscript according to the editor’s discretion.

Major Points to Address

Literature Organization. The introduction outlines the importance of highlighting the drawbacks of enzyme therapeutics and the advantages of polyelectrolyte microcapsules in addressing them. However, at the end of Page 1 through Page 2, the authors provide a laundry list of prior publications without discussion to contextualize the purpose and impact of the works. It is confusing how what has been previously done relates to the objective of the current work; if a reference does not directly affect the motivation of the authors’ work, there is no need to go into detail on what was shown. By the end of the introduction before the objective statement, the goal of this manuscript was unclear. I highly suggest restructuring the introduction in the following manner: (1) provide a broader overview of polyelectrolyte complexation, (2) introduce polymer-protein complex assemblies, (3) emphasize particularly important works in the field, and (4) state what is lacking.

Polymer + Enzyme Interactions. There are several points that need to be addressed to fully characterize the interactions. Figure 3: The authors should provide a negative control in which the amino acids are exposed as a comparison for the other samples. Figure 3: The authors should perform circular dichroism spectroscopy to evaluate the secondary structure of the protein, in order to truly conclude that “Thus, it was found that PAA affects the activity of the enzyme but does not affect its structure.” As an example, see Black, K. A.; Priftis, D.; Perry, S. L.; Yip, J.; Byun, W. Y.; Tirrell, M. “Protein Encapsulation via Polypeptide Complex Coacervation” ACS Macro Lett. 2014, 3, 1088–1091. Salt Valency Effects: The two investigated salts have differences in valency. Are the authors certain that the differences observed is a result of the mechanistic discussions provided, or if it’s more general to the fact that NaCl is a monovalent salt while (NH3)2SO4 is a divalent salt? Additional controls can directly answer this question. Encapsulation Efficiency: What is the encapsulation efficiency of ADH into the polyelectrolyte microcapsules under the various investigated conditions? Microcapsule Characterization: The authors only characterize the protein-polymer assemblies with activity assays and a single fluorescence assay. There needs to be complementary characterization tools, such as imaging (optical microscopy), scattering, or other tools (e.g., isothermal titration calorimetry, ), to fully characterize the assemblies that are formed under the investigated conditions. Ammonium Sulfate Salt Mechanism: The authors propose that sulfate anion of the salt binds to amino acid groups, thereby increasing polymer chain stiffness and reducing polymer-protein interactions. While this is plausible, without any experimental, simulation data, or even literature evidence, this is not entirely convincing from the activity-concentration. Comparison of affinity: The authors discuss Michaelis-Menten kinetics for Figure 7, but there is no information on the fit (and uncertainty) to the model and the extracted parameters. There also should be additional experiments done to back up claims made in the discussion (e.g., This effect is due to non-competitive inhibition of the enzyme).

Minor Points to Address

The following smaller points need to be clarified in the text, tables, and figures:

Page 1, Line 34: scientific work rarely uses “you” in text Page 1, Line 39: the abbreviation PCM is never formally defined Page 3, Line 84: what is the dispersity of the purchased polymers? Page 3, Line 86: what is the purity of the salts? Page 3, Line 88: did the authors check to see if the particles were net netural when selecting polymer/enzyme weight ratios? Page 4, Line 85: typo in “acohol” (alcohol) Page 4, Line 139: grammar issues throughout this paragraph Figure 1: in the x-axis, what are the commas? Is this a typo for periods? If so, the scale is misleading, since there is a break in the data at longer time points. It would be better to separate this into 2 plots. All activity-time figures: in the captions, please state what the data point and error bars represent (average and standard deviation? Of how many measurements?). Figure 4: replace all commas with periods. Page 7, Line 206: typo in “fig.” (remove) Page 9, Line 227: what does the “unacceptable” mean? There is no funding source or acknowledgement for this work. Was this omitted by mistake?

Author Response

We greatly appreciate your thoughtful comments that helped improve the manuscript. We
trust that all of your comments have been addressed accordingly in a revised manuscript.
Thank you very much for your effort.
In the following, we give a point-by-point reply to your comments:

The introduction outlines the importance of highlighting the drawbacks of enzyme therapeutics and the advantages of polyelectrolyte microcapsules in addressing them. However, at the end of Page 1 through Page 2, the authors provide a laundry list of prior publications without discussion to contextualize the purpose and impact of the works. It is confusing how what has been previously done relates to the objective of the current work; if a reference does not directly affect the motivation of the authors’ work, there is no need to go into detail on what was shown. By the end of the introduction before the objective statement, the goal of this manuscript was unclear. I highly suggest restructuring the introduction in the following manner: (1) provide a broader overview of polyelectrolyte complexation, (2) introduce polymer-protein complex assemblies, (3) emphasize particularly important works in the field, and (4) state what is lacking.

The inroduction was improved taking into account the comments of the reviewer.

Figure 3: The authors should perform circular dichroism spectroscopy to evaluate the secondary structure of the protein, in order to truly conclude that “Thus, it was found that PAA affects the activity of the enzyme but does not affect its structure.

According to the method referred to in [24], if the fluorescence of alcohol dehydrogenase does not deacreaed, then the quaternary structure is not disturbed. If the quaternary structure does not collapse, then the secondary one also, which allows us to conclude that the polymer does not affect the structure.

Figure 3: The authors should provide a negative control in which the amino acids are exposed as a comparison for the other samples.

Measurement of the fluorescence of amino acids in the presence of PAA will not show us reliable results as a control sample. ADH was used as a control sample.

Salt Valency Effects: The two investigated salts have differences in valency. Are the authors certain that the differences observed is a result of the mechanistic discussions provided, or if it’s more general to the fact that NaCl is a monovalent salt while (NH3)2SO4 is a divalent salt? Additional controls can directly answer this question.

Salts with divalent anions, except for sulfates, undergo hydrolysis, then it is not possible to conduct this experiment with reliable results. However, the mechanism of interaction of the sulfate anion with PAA that we proposed does not exclude the same result with other divalent salts.

Encapsulation Efficiency: What is the encapsulation efficiency of ADH into the polyelectrolyte microcapsules under the various investigated conditions?

In this paper, the study of encapsulated ADH was carried out only in one (selected) conditions. Encapsulation of ADH by adsorbtion method is 30%, we added that to article.

There needs to be complementary characterization tools, such as imaging (optical microscopy), scattering, or other tools (e.g., isothermal titration calorimetry, ), to fully characterize the assemblies that are formed under the investigated conditions.

We added the link to our method in the article.

Sulfate Salt Mechanism: The authors propose that sulfate anion of the salt binds to amino acid groups, thereby increasing polymer chain stiffness and reducing polymer-protein interactions. While this is plausible, without any experimental, simulation data, or even literature evidence, this is not entirely convincing from the activity-concentration.

Experimental data are shown at the line 207-210, 218-220, literature evidence are shown at the line 197-200 (reference 18) and the mechanism of it described at line 210-217 and line 220-234.

Page 3, Line 86: what is the purity of the salts?

the purity of the salts is 99%, added to the article.

Page 3, Line 88: did the authors check to see if the particles were net netural when selecting polymer/enzyme weight ratios?

The studies were conducted on a complex of polyelectrolyte and protein, without the use of particles.

Figure 1: in the x-axis, what are the commas? Is this a typo for periods? If so, the scale is misleading, since there is a break in the data at longer time points. It would be better to separate this into 2 plots. All activity-time figures: in the captions, please state what the data point and error bars represent (average and standard deviation? Of how many measurements?). Figure 4: replace all commas with periods.

We changed commas to dots.

Page 9, Line 227: what does the “unacceptable” mean?

We corrected that at the article.

There is no funding source or acknowledgement for this work. Was this omitted by mistake?

This is no mistake, we don't have any funding source.

Reviewer 2 Report

Manuscript Number: polymers-706559

Title: “Study of the Effect of Pollyallylamine on Alcoholdehydrogenase for Its Encapsulation into Polyelectrolyte Microcapsules” by Kim et al.

Reviewer’s comments:

I have a few questions/suggestions for the current version of the manuscript.

The title is not clear to me. Please revise the title so that it clearly reflects the main theme of the research work. In the current title, Alcoholdehydrogenase à Alcohol Dehydrogenase To meet the quality of the journal, editorial proofreading throughout the manuscript is strongly recommended. Please be aware that there seems to be several typographical errors, and problems in word usage and sentence construction in this version of the manuscript. Each abbreviation should be explained at its first appearance in the main text. The abbreviation “PMC” appears in its first time without mentioning its complete meaning. Statistical analysis should be performed and its description should be included in the materials and methods section. In the introduction section, the parts including the paragraphs ranging from lines 47-79 should be merged into one paragraph. It was stated that “This study allows create a diagnostic system of multiple-use based on polyelectrolyte microcapsules for measurement concentration of ethanol.” without indicating how. I think Figure 2 is not necessary. What is the emission wavelength used in the “Determination of fluorescence intensity of ADH” and also Figure 3? Why chose sodium chloride and ammonium sulfate. In lines 181-182, “Such a drop-in protein activity is associated with a decrease in its solubility due to salting out and dehydration with salt.”: Please include the relevant citations/references to support this passage. Why was Figure 6 included in this manuscript? I think Figure 6 is not necessary. I am assuming that the kinetic data obtained were further fitted against the kinetic models (Michaelis–Menten model and/or its modified version with and without inhibition). However, no information about the equations and associated parameters was provided. What is the definition of affinity constant? How well did the model fit the data? Please show R-squares. The readability of this manuscript should be greatly improved. It is hard to follow in some parts. Comparison and discussion of the results obtained here and previously reported are lacking.

Author Response

We greatly appreciate your thoughtful comments that helped improve the manuscript. We
trust that all of your comments have been addressed accordingly in a revised manuscript.
Thank you very much for your effort.
In the following, we give a point-by-point reply to your comments:

 The abbreviation “PMC” appears in its first time without mentioning its complete meaning. In the introduction section, the parts including the paragraphs ranging from lines 47-79 should be merged into one paragraph. Statistical analysis should be performed and its description should be included in the materials and methods section. It was stated that “This study allows create a diagnostic system of multiple-use based on polyelectrolyte microcapsules for measurement concentration of ethanol.” without indicating how. What is the emission wavelength used in the “Determination of fluorescence intensity of ADH” and also Figure 3?

We corrected that at the article.

I think Figure 2 is not necessary.

We deleted that figure

Why chose sodium chloride and ammonium sulfate.

Ammonium sulfate and sodium chloride salts were used by way of example of bivalent and monovalent salts. We added that at the article.

In lines 181-182, “Such a drop-in protein activity is associated with a decrease in its solubility due to salting out and dehydration with salt.”: Please include the relevant citations/references to support this passage.

We added that at the article.

Why was Figure 6 included in this manuscript? I think Figure 6 is not necessary.

We deleted that figure

What is the definition of affinity constant?

It is a dissociation constant used in receptor binding and enzyme inhibition. The higher the affinity constant, the greater the ligand affinity for the receptor.

I am assuming that the kinetic data obtained were further fitted against the kinetic models (Michaelis–Menten model and/or its modified version with and without inhibition). However, no information about the equations and associated parameters was provided.

Kinetic data were not compared with the kinetic model; Km was obtained to understand the level of enzyme affinity for the substrate.

Reviewer 3 Report

Reviewing the manuscript “Study of the Effect of Pollyallylamine on Alcoholdehydrogenase for Its Encapsulation into Polyelectrolyte Microcapsules” submitted to “Polymersl” for publication revealed this work interesting and fit well with in the scope of this journal. In this experimental study, authors have investigate the effect of polyallylamine (PAA) on the structure and catalytic characteristics of alcohol dehydrogenase (ADH).

This is a well-designed study and the manuscript fits well within the scope of the journal; it needs some improvements; there are a few suggestions that authors may consider to improve it further:

Title: I suggest to remove the words “Study of the” and title should be “Effects of Pollyallylamine on Alcoholdehydrogenase for Its Encapsulation into Polyelectrolyte Microcapsules”

Abstract: is unstructured and presenting key information; however, there is no much description of methodology (only techniques used were mentioned). Authors are requested to include some more details of methodology in the abstract section.

The use of English language is reasonable, however, there are a number of punctuation and grammatical errors; that should be corrected and rephrased using academic English for a better flow of text for reader. I could see a few in abstract and introduction too. Authors should proofread to eradicate typos and grammatical errors.

Methods and results: are described in well details. Although authors mentioned “significance” in results however there are no details of statistical analysis in the methods section. Authors should include data analysis or justify.

What are the limitations of the study? Please detail before conclusion if there were any limitations.

The conclusions section is very much detailed: authors are advised to remove repetition and condense conclusion section.

line 256-57 are grammatically incorrect, please check and rephrase.

Author Response

We greatly appreciate your thoughtful comments that helped improve the manuscript. We
trust that all of your comments have been addressed accordingly in a revised manuscript.
Thank you very much for your effort.
In the following, we give a point-by-point reply to your comments:

I suggest to remove the words “Study of the” and title should be “Effects of Pollyallylamine on Alcoholdehydrogenase for Its Encapsulation into Polyelectrolyte Microcapsules”

We corrected that at the article.

Authors are requested to include some more details of methodology in the abstract section.

We indicated in the abstract the main aspects of the methodology: "For this research, we used methods of stationary kinetics and fluorescence spectroscopy." A more detailed description would greatly increase the size of the abstract. We indicated in the abstract the main aspects of the methodology:

The use of English language is reasonable, however, there are a number of punctuation and grammatical errors; that should be corrected and rephrased using academic English for a better flow of text for the reader. I could see a few in abstract and introduction too. Authors should proofread to eradicate typos and grammatical errors. Although the authors mentioned “significance” in results however there are no details of statistical analysis in the methods section. Authors should include data analysis or justify. What are the limitations of the study? line 256-57 are grammatically incorrect, please check and rephrase.

We corrected that at the article.

line 256-57 are grammatically incorrect, please check and rephrase.

We corrected that at the article.

What are the limitations of the study?

We thank you for your question, but what does it mean "limitations of the study"? It will be helpful if you explain to us what does it mean. 

Round 2

Reviewer 1 Report

In the resubmitted manuscript, while Kim and coworkers have revised their work on the effect of polyallylamine (PAA) on alcohol dehydrogenase (ADH) structure and activity.

While many of the changes have improved the manuscript, overall the issues raised in the previous report are not adequately addressed. Two of the other reviewers also bring up an excellent point about conducting statistical analysis on the collected data, which was not done. The new introduction also presents new issues that need additional revisions. See below for details. There are significant issues left to do, and in its current state, I cannot recommend this manuscript for publication in Polymers.

  • In the revised Introduction, there are grammar issues and typos (e.g., Line 47), as well as missing/mis-ordered citations (e.g., Lines 48, 49). Overall, the narrative is still lacking, with the Introduction reading as a Review article rather than outlining the necessary background that motivates the novelty of the authors' work.
  • For Figure 3, the cited work described a study the kinetics of acid denaturation of horse liver alcohol dehydrogenase. The authors do not make the general conclusion that "if the fluorescence of alcohol dehydrogenase does not decrease, then the quaternary structure is not disturbed" as the authors assert. To fully describe the mechanism and support the data, additional complementary experiments are needed; otherwise, this is just speculation.
  • Other sulfate salts can address the monovalent versus divalent salt question. An expanded library of controls is needed to justify the conclusions based on just NaCl and (NH3)2SO4.

Author Response

We greatly appreciate your thoughtful comments that helped improve the manuscript. We trust that all of your comments have been addressed accordingly in a revised manuscript.

Thank you very much for your effort.

In the following, we give a point-by-point reply to your comments:

Two of the other reviewers also bring up an excellent point about conducting statistical analysis on the collected data, which was not done. 

Chapter about statistical analysis was added to manuscript. 

For Figure 3, the cited work described a study the kinetics of acid denaturation of horse liver alcohol dehydrogenase. The authors do not make the general conclusion that "if the fluorescence of alcohol dehydrogenase does not decrease, then the quaternary structure is not disturbed" as the authors assert. To fully describe the mechanism and support the data, additional complementary experiments are needed; otherwise, this is just speculation.

The study of the fluorescence intensity of a protein is widely used to study its structure. (Maurice R. Eftink, Fluorescence Techniques for Studying Protein Structure, Methods of Biochemical Analysis, 1991;)

Example, Makoto Yoshimoto et al. at chapter 3.4.3. demonstrated the fluorescence intensity of catalase is significantly decreased after the heat treatment and make conclusion, that these results indicate that a conformationally changed catalase molecules induced by the heat treatment. (Yoshimoto M, Sakamoto H, Yoshimoto N, Kuboi R, Nakao K (2007) Stabilization of quaternary structure and activity of bovine liver catalase through encapsulation in liposomes. Enzyme Microb Technol 41:849–858)

The analogical results and conclusion was make about ascorbate oxidase in the article of Mei G. et al.(Mei, G., Di Venere, A., Buganza, M., Vecchini, P., Rosato, N., & Finazzi-Agro’, A. (1997). Role of Quaternary Structure in the Stability of Dimeric Proteins: The Case of Ascorbate Oxidase. Biochemistry, 36(36), 10917–10922. doi:10.1021/bi970614p)

Also, we added a citation to methodical materials, which demonstrated fluorescence intensity. “Fluorescence Spectroscopy: A tool for Protein folding/unfolding Study.” (2007). Available online: https://www.semanticscholar.org/paper/Fluorescence-Spectroscopy%3A-A-tool-for-Protein-Study/9cbe89466a4e30fb925fa7fc51258d06f9da4740#paper-header

Other sulfate salts can address the monovalent versus divalent salt question. An expanded library of controls is needed to justify the conclusions based on just NaCl and (NH3)2SO4.

In that article, we study the effect of polycation (PAA) to enzyme structure, the anion of sulfate ammonium and its valence effect to that process. We demonstrated the lack of influence of cation in the article (18), reference was added in the section - Bibliography. The study of the influence of sulfate anion with another cations will not answer to the question about valence influence. 

We postulated lack of polyelectrolyte influence to activity of enzyme in the presence of sulfate ammonium. That effect is associated with changing of stiffness polyelectrolyte chain because sulfate anion connecting with two amino groups of PAA. Since we look at the interaction of sulfate anion with polycation (PAA), the addition of other cations will not give an answer about the influence of salt valence.

Reviewer 2 Report

1. Original Question: What is the definition of affinity constant?

Authors' answer is: It is a dissociation constant used in receptor binding and enzyme inhibition. The higher the affinity constant, the greater the ligand affinity for the receptor.

Are you sure affinity constant is the same as the dissociation constant? What I see in the manuscript is the Michaelis constant "Km". I think authors are confused about the definitions on affinity constant, dissociation constant, and Michaelis constant (Km). Please make a clear definition on each term.

2. Original Question: I am assuming that the kinetic data obtained were further fitted against the kinetic models (Michaelis–Menten model and/or its modified version with and without inhibition). However, no information about the equations and associated parameters was provided.

Authors' Answer: Kinetic data were not compared with the kinetic model; Km was obtained to understand the level of enzyme affinity for the substrate.

In that case, how did you come up with the value of Km without performing fitting process

3. The introduction section should be re-organized. Several paragraphs can be combined accordingly.

4. Statistical analysis procedure should be performed on the data and statistical analysis section should also be included in the manuscript.

5. The quality of figures should be greatly improved.

6. The presentation of units should be consistent. For example, mg or Mg? mL or ML? ml or mL?

Author Response

We greatly appreciate your thoughtful comments that helped improve the manuscript. We trust that all of your comments have been addressed accordingly in a revised manuscript.

Thank you very much for your effort.

In the following, we give a point-by-point reply to your comments:

  1. Original Question: What is the definition of affinity constant?

Authors' answer is: It is a dissociation constant used in receptor binding and enzyme inhibition. The higher the affinity constant, the greater the ligand affinity for the receptor.

Are you sure affinity constant is the same as the dissociation constant? What I see in the manuscript is the Michaelis constant "Km". I think authors are confused about the definitions on affinity constant, dissociation constant, and Michaelis constant (Km). Please make a clear definition on each term.

  1. We changed affinity constant to Km in text because that kind of terminology can confuse the readers.
  1. Original Question: I am assuming that the kinetic data obtained were further fitted against the kinetic models (Michaelis–Menten model and/or its modified version with and without inhibition). However, no information about the equations and associated parameters was provided. Authors' Answer: Kinetic data were not compared with the kinetic model; Km was obtained to understand the level of enzyme affinity for the substrate. In that case, how did you come up with the value of Km without performing fitting process
  1. Km is numerically equal to the substrate concentration with which reaction rate (V) equal to the half of maximum reaction rate (Vm). (Michaelis L., Menten M. L. Die kinetik der invertinwirkung //Biochem. z. — 1913. — Т. 49. — â„–. 333—369. — С. 352.)
  1. The introduction section should be re-organized. Several paragraphs can be combined accordingly.

The introduction was re-organized.

  1. Statistical analysis procedure should be performed on the data and statistical analysis section should also be included in the manuscript.

The chapter about statistical analysis was added to manuscript and to the text.

  1. The quality of figures should be greatly improved.

The quality of figures was greatly improved

  1. The presentation of units should be consistent. For example, mg or Mg? mL or ML? ml or mL?

The presentation of units corrected.

Reviewer 3 Report

Many thanks for responding to the comments; 

the limitations of the study should be added; for instance, what is missing in this study design that may or may not be improved in further studies. what are are the potential factors that may affect the validity of the results/data etc.

Author Response

We greatly appreciate your thoughtful comments that helped improve the manuscript. We trust that all of your comments have been addressed accordingly in a revised manuscript.

Thank you very much for your effort.

In the following, we give a point-by-point reply to your comments:

The limitations of the study should be added; for instance, what is missing in this study design that may or may not be improved in further studies. what are are the potential factors that may affect the validity of the results/data etc.

The limitations of the study connected with the inability to experiment with other divalent salts. Salts with divalent anions, except for sulfates, undergo hydrolysis, then it is not possible to conduct this experiment with reliable results. However, the mechanism of interaction of the sulfate anion with PAA that we proposed does not exclude the same result with other divalent salts.

Round 3

Reviewer 1 Report

Kim and coworkers have provided sufficient changes to address many of the reviewers' concerns. There are only a few last minor points in the tracked changes, listed below. Once these have been resolved, I support publication to Polymers.

Introduction

  • Line 34: The abbreviation "(PCM)" should be "(PCMs)" because the word microcapsules is plural.
  • Line 51: There are no citations to a wide number of recent reviews for the topic: "There are a number of works in the literature that solve similar problems and show various options for complexation and the effect of polyelectrolyte on protein." This would be helpful to provide broad references for nonexperts. Some suggestions are:
    • Horn, J.; Kapelner, R.; Obermeyer, A. Macro- and Microphase Separated Protein-Polyelectrolyte Complexes: Design Parameters and Current Progress. Polymers 2019, 11 (4), 578.
    • Acar, H.; Ting, J. M.; Srivastava, S.; LaBelle, J. L.; Tirrell, M. V. Molecular Engineering Solutions for Therapeutic Peptide Delivery. Chem. Soc. Rev. 2017, 46, 6553–6569.
    • de Kruif, C. G.; Weinbreck, F.; de Vries, R. Complex Coacervation of Proteins and Anionic Polysaccharides. Curr. Opin. Colloid Interface Sci. 2004, 9 (5), 340–349.
  • For all statistical tests, the authors report the average and standard deviation, but the number of repeats (N) is not provided. Please state whether these measurements were done in duplicate, triplicate, etc.
  • Line 281: "The limitations of the study" is not a complete sentence and serves as a subheader. Please either re-write this sentence or change the heading. It would also be beneficial to include some ideas and discussions that perhaps the authors are currently doing to address these limitations. 

Author Response

We greatly appreciate your thoughtful comments that helped improve the manuscript. We trust that all of your comments have been addressed accordingly in a revised manuscript.

Thank you very much for your effort.

In the following, we give a point-by-point reply to your comments:

  • Line 34: The abbreviation "(PCM)" should be "(PCMs)" because the word microcapsules is plural.

The abbreviation was corrected. 

  • Line 51: There are no citations to a wide number of recent reviews for the topic: "There are a number of works in the literature that solve similar problems and show various options for complexation and the effect of polyelectrolyte on protein." This would be helpful to provide broad references for nonexperts. 

The references were added. 

  • For all statistical tests, the authors report the average and standard deviation, but the number of repeats (N) is not provided. Please state whether these measurements were done in duplicate, triplicate, etc.

The number of repeats was added. 

  • Line 281: "The limitations of the study" is not a complete sentence and serves as a subheader. Please either re-write this sentence or change the heading. It would also be beneficial to include some ideas and discussions that perhaps the authors are currently doing to address these limitations. 

The limitation of the study was corrected.